# Adenine methylation is very scarce in the *Drosophila* genome and not erased by the ten-eleven translocation dioxygenase

Manon Boulet[1†], Guerric Gilbert[1†], Yoan Renaud[1†], Martina Schmidt-Dengler[2], Emilie Plantié[1], Romane Bertrand[1], Xinsheng Nan[3], Tomasz Jurkowski[3], Mark Helm[2], Laurence Vandel[1*], Lucas Waltzer[1*]

[1]Université Clermont Auvergne, CNRS, INSERM, iGReD, Clermont-Ferrand, France; [2]Institute of Pharmaceutical and Biomedical Sciences, Johannes Gutenberg-Universität, Mainz, Germany; [3]School of Biosciences, Cardiff University, Cardiff, United Kingdom

**Abstract** N6-methyladenine (6mA) DNA modification has recently been described in metazoans, including in *Drosophila*, for which the erasure of this epigenetic mark has been ascribed to the ten-eleven translocation (TET) enzyme. Here, we re-evaluated 6mA presence and TET impact on the *Drosophila* genome. Using axenic or conventional breeding conditions, we found traces of 6mA by LC-MS/MS and no significant increase in 6mA levels in the absence of TET, suggesting that this modification is present at very low levels in the *Drosophila* genome but not regulated by TET. Consistent with this latter hypothesis, further molecular and genetic analyses showed that TET does not demethylate 6mA but acts essentially in an enzymatic-independent manner. Our results call for further caution concerning the role and regulation of 6mA DNA modification in metazoans and underline the importance of TET non-enzymatic activity for fly development.

**\*For correspondence:**
laurence.vandel@uca.fr (LV);
lucas.waltzer@uca.fr (LW)

[†]co-first authors

**Competing interest:** The authors declare that no competing interests exist.

## eLife assessment

This study investigates the presence of DNA adenine methylation (6mA) and the associated function of TET enzyme, a DNA methylation mark eraser, in *Drosophila*. The study presents **valuable** findings on the scarcity of 6mA in the *Drosophila* genome and challenges previous findings regarding the role of TET in 6mA modification. The evidence supporting the claims is **solid**, and the paper has the potential to stimulate re-evaluations of the significance and regulatory mechanisms of 6mA DNA modifications in *Drosophila*.

## Introduction

Until recently, N[6]-methyl-2'-deoxyadenosine (also called N6-methyladenine or 6mA) was considered to be essentially restricted to the genome of prokaryotes, where this modification plays a well-established role in the restriction-modification system and other processes such as DNA replication or transcription (*Sánchez-Romero and Casadesús, 2020*; *Wion and Casadesús, 2006*). Since 2015, several reports detected the presence of 6mA in the DNA of different eukaryotic organisms (*Alderman and Xiao, 2019*; *Boulias and Greer, 2022*), including in metazoans (*Greer et al., 2015*; *Koziol et al., 2016*; *Liu et al., 2016*; *Wu et al., 2016*; *Xiao et al., 2018*; *Xie et al., 2018*; *Zhang et al., 2015*). Although a small fraction of all adenines seems methylated at N6 position (from 0.4 to 0.0001% or below), it was proposed that this modification participates in eukaryotic genome regulation (*Wu, 2020*). Yet, the significance of 6mA in eukaryotes and the enzymes involved in its metabolism remain controversial

with several studies questioning the existence and/or the level of this modification, particularly in metazoans (*Douvlataniotis et al., 2020*; *Kong et al., 2022*; *Liu et al., 2021*; *Musheev et al., 2020*; *O'Brown et al., 2019*; *Schiffers et al., 2017*). Part of the controversy stems from the technologies used to detect 6mA (*Boulias and Greer, 2022*; *Li et al., 2021*). Notably, antibody-based techniques such as dot blot or DNA immunoprecipitation followed by sequencing (DIP-seq) have been particularly called into question to study low levels of 6mA (*Bochtler and Fernandes, 2021*; *Douvlataniotis et al., 2020*; *Lentini et al., 2018*). If liquid chromatography coupled with tandem mass spectrometry (LC-MS/MS) provides a sensitive method to identify 6mA and measure its absolute levels unambiguously, bacterial contaminations can affect the results (*Douvlataniotis et al., 2020*; *Kong et al., 2022*; *O'Brown et al., 2019*). Finally, single-molecule real-time sequencing (SMRT-seq) can detect 6mA presence (and location) on genomic DNA but is prone to give rise to a high false discovery rate when 6mA is rare (*Douvlataniotis et al., 2020*; *O'Brown et al., 2019*; *Zhu et al., 2018*).

Notwithstanding, 6mA presence appears strongly supported in *Drosophila* genome (*He et al., 2019*; *Ismail et al., 2019*; *Shah et al., 2019*; *Yao et al., 2018*; *Ye et al., 2017*; *Zhang et al., 2015*), where this modification was described to be associated with transposable element silencing and activation of gene transcription (*He et al., 2019*; *Yao et al., 2018*). Unexpectedly, 6mA demethylation in *Drosophila* was attributed to the TET enzyme (*Zhang et al., 2015*), a member of the 5-methylcytosine (5mC) dioxygenase family (*Iyer et al., 2009*).

Here, we re-evaluated 6mA levels in *Drosophila* and reassessed the impact of TET on this mark. Using LC-MS/MS, we show that 6mA is present at very low levels in *Drosophila* in axenic conditions and that the absence of TET does not lead to any consistent increase in 6mA levels in the larval central nervous system, nor in the whole larva, the embryo or the adult brain. Furthermore, our genetic and molecular analyses suggest that TET is not involved in 6mA demethylation and that its function during *Drosophila* development is largely catalytic-independent.

## Results and discussion

Previously reported levels of 6mA measured by LC/MS-MS in *Drosophila* ranged from 0.07 to 0.0006% (6mA/A), with the highest levels in the early embryo (*Yao et al., 2018*; *Zhang et al., 2015*). However, a recent study reported much lower levels (0.0002%) even in early embryos and showed that initially reported 'high' levels of 6mA were likely due to bacterial contaminations (*Kong et al., 2022*). Indeed, the contamination of genomic DNA (gDNA) by bacterial DNA is a major confounding factor for LC-MS/MS experiments (*Douvlataniotis et al., 2020*; *Kong et al., 2022*; *O'Brown et al., 2019*). Moreover, the presence of intracellular bacteria can also be a source of 6mA (*Douvlataniotis et al., 2020*). Along that line, it is worth noting that the genome of *Wolbachia*, a frequent endosymbiont in *Drosophila*, codes for DNA Adenine methyltransferases (*Saridaki et al., 2011*). In addition, 6mA derived from exogenous sources might be incorporated into gDNA *via* the salvage pathway (*Musheev et al., 2020*), and independently of autonomously-directed adenine methylation (*O'Brown et al., 2019*; *Schiffers et al., 2017*). To exclude these possible sources of contamination, we generated germ-free *Drosophila* and reared the larvae on chemically-defined ('holidic') food devoid of exogenous DNA contribution (see Methods). The absence of exogenous or endosymbiotic bacteria in the resulting flies was confirmed by PCR (*Figure 1A*). In these conditions, we observed 0.00025% of 6mA in whole larvae (*Figure 1B*) and 0.0005% in the larval central nervous system (CNS) (*Figure 1C*). Noteworthy, this corresponds to around 200–400 methylated adenines per haplogenome. In addition, similar levels of 6mA were measured in the CNS when non-axenic larvae were reared on classic medium without antibiotic treatment (*Figure 1C*), suggesting that contamination by exogenous sources is not a major problem in this tissue. These results confirm the presence of a very small fraction of methylated adenines in *Drosophila* DNA.

Changes in 6mA levels following genetic manipulations of putative adenine methylases or demethylases have brought further credence to the existence and role of this modification in metazoans (*Boulias and Greer, 2022*). In *Drosophila*, the absence of TET (also called DMAD, for DNA Methyl Adenine Demethylase) was associated with a strong increase in 6mA levels in the embryo or adult ovary and brain (*Yao et al., 2018*; *Zhang et al., 2015*). In addition, in vitro experiments suggested that *Drosophila* TET can demethylate 6mA (*Yao et al., 2018*; *Zhang et al., 2015*). However, that TET mediates 6mA oxidation is at odds with the well-characterized function of this family of enzymes in 5mC oxidation in metazoans (*Lio et al., 2020*). Moreover, the other enzymes involved in methyladenine

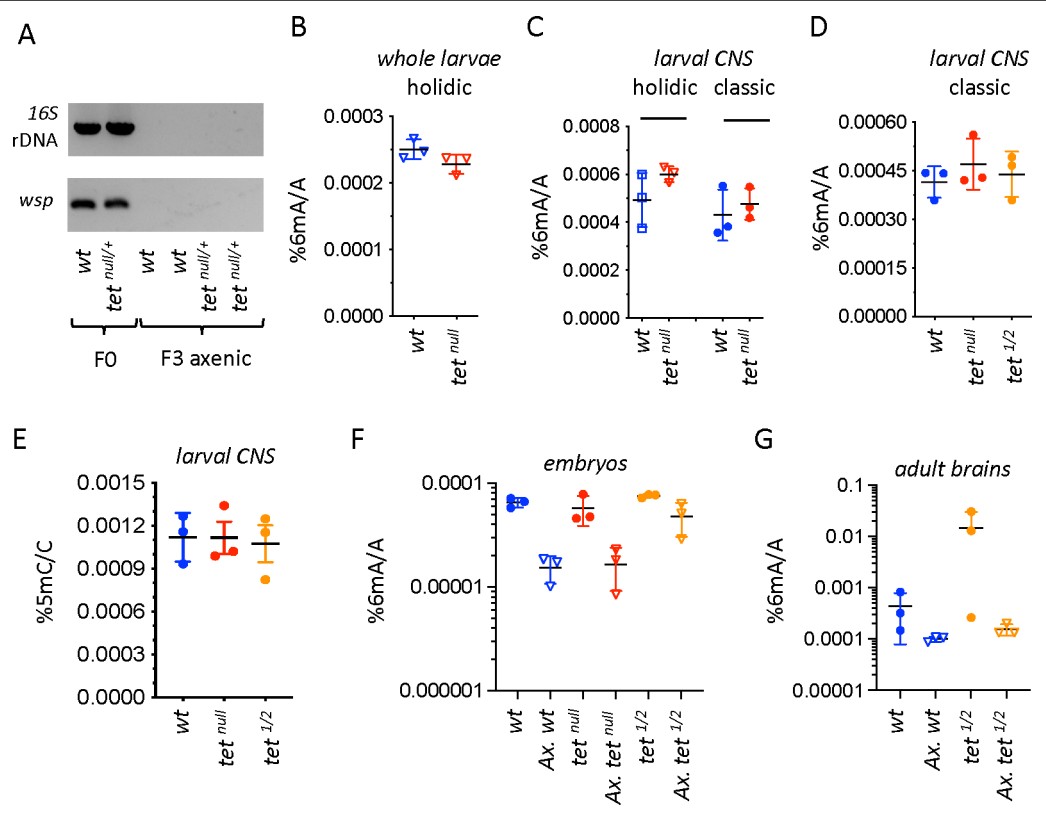

**Figure 1.** N6-methyladenine (6mA) levels in axenic *Drosophila* larvae are very low and not affected by ten-eleven translocation (TET) loss. (**A**) The presence of bacterial contaminations in wild-type (*wt*) and *tet*[null/+] adult flies was checked by PCR using universal primers against bacterial 16 S rDNA and against the endosymbiotic bacteria gene *wsp*. PCRs were performed on DNA from parental (F0) flies and after three generations of breeding in axenic conditions (F3). (**B–D**) 6mA levels were measured by liquid chromatography with tandem mass spectrometry (LC-MS/MS) in genomic DNA (gDNA) from whole larvae (**B**) or dissected central nervous system (CNS) (**C, D**) generated from axenic flies reared on holidic medium (**B, C**) or conventional flies reared on classic medium (**C, D**). (**E**) 5-methylcytosine (5mC) levels were measured by LC-MS/MS in gDNA from dissected CNS from flies reared on classic medium. (**F, G**) 6mA levels were measured by LC-MS/MS in embryos (**F**) and dissected adult brains (**G**) collected from crosses with conventional or axenic (*Ax.*) individuals raised on classic fly medium supplemented (*Ax.*) or not with antibiotics. *wt*: wild-type (*w*[1118]); *tet*[null]: *tet*[null/null]; *tet*[1/2]: *tet*[DMAD1/DMAD2]. Filled circles: conventional flies; open triangles: axenic flies. Individual values, means and standard deviations are plotted. No statistically significant differences were observed between *wt* and *tet* mutant samples (Mann-Whitney test).

The online version of this article includes the following source data and figure supplement(s) for figure 1:

**Source data 1.** Original file of the raw gels of PCR analyses for bacterial contaminations.

**Source data 2.** PDF containing *Figure 1A* and raw gel pictures with relevant labels.

**Figure supplement 1.** Detection of bacterial contamination in parental stocks and after three generations of breeding in axenic conditions.

**Figure supplement 1—source data 1.** Original file of the raw gels of PCR analyses for bacterial contaminations.

**Figure supplement 1—source data 2.** PDF containing *Figure 1—figure supplement 1a* and raw gel pictures with relevant labels.

oxidation/demethylation belong to the AlkB family (*Boulias and Greer, 2022*; *Xu and Bochtler, 2020*), which is related to, but distinct from the TET family (*Iyer et al., 2009*; *Jia et al., 2017*). Indeed, conserved residues involved in TET 5mC recognition differ from those found within the AlkB family and may not be able to accommodate a purine residue instead of a pyrimidine (*Hu et al., 2013*; *Iyer et al., 2009*; *Parker et al., 2019*; *Xu and Bochtler, 2020*). Nevertheless, the fact that the *Drosophila* genome presents extremely low levels of 5mC and does not code for any 5mC DNA methyltransferase (*Iyer et al., 2011*; *Krauss and Reuter, 2011*) prompted the idea that TET could catalyze other

forms of DNA modifications and notably 6mA oxidation/demethylation in the absence of its canonical substrate (*Zhang et al., 2015*). As *tet* was shown to be highly expressed in the larval CNS (*Delatte et al., 2016*; *Wang et al., 2018*), we first focused on its impact on 6mA in this tissue. Yet, we found that the levels of 6mA measured by LC-MS/MS in the absence of TET *tet*$^{null}$, an allele that abolishes *tet* transcription (*Delatte et al., 2016*), were similar to wild-type in the larval CNS using either axenic flies raised on holidic medium or non-axenic flies raised on classic medium (i.e. conventional conditions) (*Figure 1C*). Of note, TET loss did not show any impact on 6mA level either in whole larvae (*Figure 1B*). As previous experiments showing an increase in 6mA in the absence of TET were performed under conventional conditions with *tet*$^{DMAD1}$/*tet*$^{DMAD2}$ mutant alleles (*Yao et al., 2018*; *Zhang et al., 2015*), which introduced a premature stop codon in *tet* open reading frame before its catalytic domain (*Zhang et al., 2015*), we repeated the analyses with this allelic combination. Yet, we did not find any increase of 6mA levels in the larval CNS in this setting either (*Figure 1D*). Moreover, consistent with previous results showing that TET does not control 5mC oxidation in *Drosophila* (*Zhang et al., 2015*), we observed very low levels of 5mC (around 0.001%) in the larval CNS and no increase upon TET loss (*Figure 1E*). In addition, the first product of 5mC oxidation, 5hmC (5-hydroxymethylcytosine), was below the detection limit (0.00001%). These results suggest that TET is not involved in 6mA (or 5mC) demethylation in the *Drosophila* larval CNS.

To test whether the lack of impact of TET on 6mA levels that we observe here contrary to previous studies (*Yao et al., 2018*; *Zhang et al., 2015*), could be due to a tissue-specific effect or to breeding conditions, we assessed 6mA levels in embryos and adult brains using conventional or germ-free flies (see Methods). As *tet*$^{null}$ homozygote mutation is pupal lethal (*Zhang et al., 2015*), only *tet*$^{DMAD1/DMAD2}$ could be used for adult brains. The presence/absence of bacterial contaminants in conventional *versus* germ-free stocks was validated by PCR (*Figure 1—figure supplement 1*). Moreover, by performing gDNA sequencing, we found around 2% of bacterial DNA contaminant in gDNA in conventional flies *versus* less than 0.003% in germ-free flies (*Figure 1—figure supplement 1*). Hence, possible traces of bacterial contaminations in axenic samples should have a negligible impact on LC-MS/MS measurements. LC-MS/MS analyses showed that 6mA levels were higher in embryos (*Figure 1F*) or adult brains (*Figure 1G*) using conventional flies as compared to their germ-free siblings. They were also more variable across samples in non-axenic conditions. It is thus likely that 6mA levels measured in non-axenic conditions do not solely reflect endogenous 6mA in the *Drosophila* genome and variations between genotypes should be interpreted with caution. Still, we did not observe any significant increase in 6mA levels in the absence of TET in non-axenic conditions (*Figure 1F and G*). Importantly, the same observation was made in axenic conditions (*Figure 1F and G*). Of note, the apparent increase in 6mA levels in *tet*$^{DMAD1/DMAD2}$ axenic embryos was not reproduced in *tet*$^{null}$ embryos, suggesting that it does not simply reflect *tet* loss-of-function, and it was not statistically significant as compared to wild-type levels (Welch two sample t-test: p=0.075). Altogether, we did not find consistent evidence that TET loss caused an increase in 6mA levels in embryos, whole larvae, larval CNS, or adult brain.

As an alternate method to study TET impact on 6mA, we used SMRT sequencing (*Boulias and Greer, 2022*). We generated SMRT-seq data on CNS gDNA from three biological replicates of wild-type and *tet*$^{null}$ larvae. As genome coverage is an important parameter to analyze SMRT-seq data, we first merged the three replicates to increase read density. In the resulting fusion datasets, around 95% of all the adenines were covered at least 25 x (*Figure 2—figure supplement 1* and *Supplementary file 1*). The detection of 6mA by SMRT-seq is based on a modification quality value (mQV or QV), reflecting the consistency by which a specific modification is observed at a given position in a subread. Using standard parameters (coverage ≥25 x and QV ≥20), we found respectively, 0.158% and 0.172% of potential 6mA in the CNS of wild-type and *tet*$^{null}$ larvae (*Figure 2A*, *Supplementary file 2*), which is much more than expected based on LC-MS/MS measurements. However, when we considered the methylation status of these adenines in the individual samples, only 2.6% were labeled as 6mA in all three replicates and 13.7% in at least two replicates (*Figure 2B* and *Supplementary file 3*), consistent with the idea that SMRT-seq gives a high rate of false positives in organisms containing low 6mA levels (*Douvlataniotis et al., 2020*; *Kong et al., 2023*; *Schadt et al., 2013*; *Zhu et al., 2018*). Interestingly, increasing the QV strongly ameliorated the fraction of replication and drastically reduced the proportion of potential 6mA both in wild-type and *tet*$^{null}$ datasets, whereas increasing the coverage had little effect (*Figure 2A and B*, *Figure 2—figure supplement 2 and 3*, and *Supplementary file 3*). We thus analyzed SMRT-seq results from wild-type and *tet*$^{null}$ replicates using a coverage ≥25 x and increasing

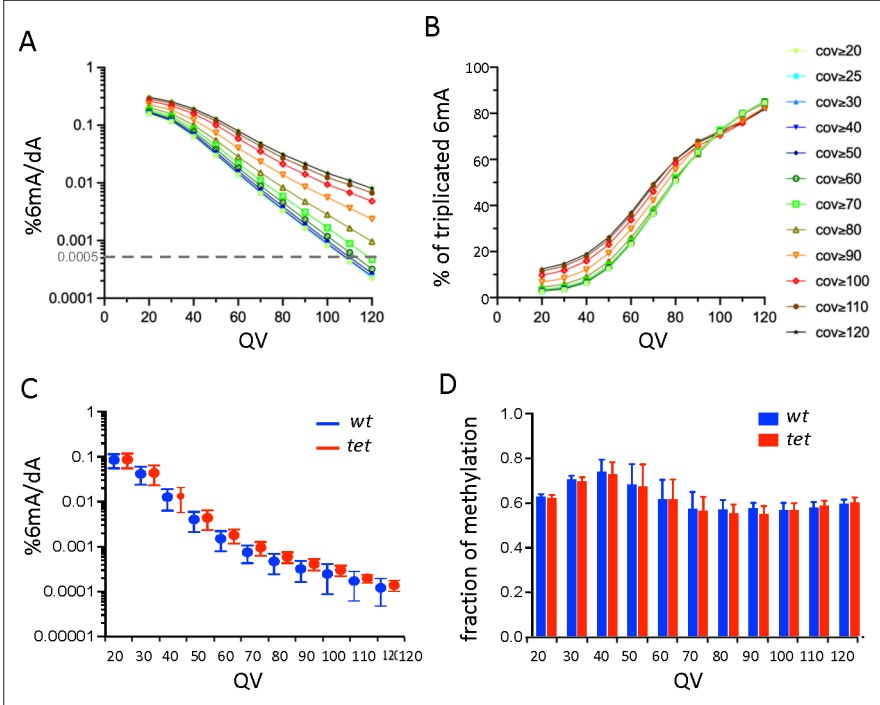

**Figure 2.** Single-molecule real-time sequencing (SMRT-seq) analysis of larval central nervous system (CNS) genomic DNA (gDNA) does not reveal an increase in N6-methyladenine (6mA) in the absence of ten-eleven translocation (TET). (**A**) Percentage of adenines identified as 6mA in the wild-type fusion dataset depending on the quality value (QV) and coverage values used for 6mA selection. The dashed gray line indicates the level of 6mA measured by liquid chromatography with tandem mass spectrometry (LC-MS/MS) (0.0005%). (**B**) Influence of the QV and the coverage values on the proportion of 6mA identified in the wild-type fusion dataset and in the three original samples. (**C**) Percentage of adenines covered at least 25 x and identified as 6mA in each of the three wild-type (*wt*) or *tet null* (*tet*) datasets depending on the QV. Means and standard deviations are represented. (**D**) Fraction of methylation in *wt* or *tet null* datasets depending on the QV (coverage ≥25 x). Means and standard deviations are represented.

The online version of this article includes the following figure supplement(s) for figure 2:

**Figure supplement 1.** Proportion of the *Drosophila* genome covered in *wild-type* (*wt*) and *tet*[null] (*tet*) single-molecule real-time sequencing (SMRT-seq) fusion datasets according to coverage density.

**Figure supplement 2.** Influence of the quality value (QV) and the coverage values on the proportion of N6-methyladenine (6mA) identified in the wild-type (*wt)* fusion dataset and in the three original samples.

**Figure supplement 3.** Influence of the quality and coverage values on SMRT-seq based detection of 6mA in the CNS of *tet*[null] larvae.

QV. However, we did not observe any significant differences either in the percentage of 6mA/A or in the fraction of methylation of these potential 6mA even with stringent QV values (*Figure 2C and D* and *Supplementary file 4*). Hence, while SMRT-seq data are noisy, as cautioned in previous studies, they did not reveal an increase in 6mA levels in the absence of TET.

To directly test whether TET demethylates 6mA, we then assessed its activity in vitro. Accordingly, the recombinant catalytic domain of *Drosophila* TET was incubated with double-stranded DNA (dsDNA) containing either a 5mC or a 6mA modification and the level of modified bases was quantified by LC-MS/MS at different time points. Under our experimental conditions, 5mC levels were drastically reduced in 1 min with the concomitant appearance of 5mC oxidation products (5hmC and 5fC) (*Figure 3A*). In sharp contrast, the level of 6mA remained constant even after 30 min of incubation (*Figure 3B*). Hence, contrary to 5mC, 6mA does not seem to be a good substrate for *Drosophila* TET in vitro. Of note, only traces of 5mC, 5hmC, 5fC, or 6mA were observed when recombinant TET was incubated with non-modified dsDNA, indicating that the levels measured in the presence of modified dsDNA were not due to contaminations (*Figure 3—figure supplement 1*). Our results contrast with previous reports showing that recombinant *Drosophila* TET demethylates 6mA on dsDNA in vitro

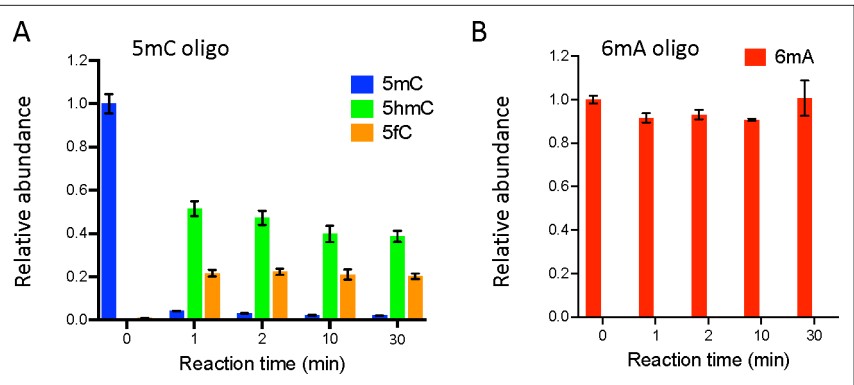

**Figure 3.** Ten-eleven translocation (TET) does not oxidize N6-methyladenine (6mA). (**A, B**) In vitro assays showing TET activity profile on 5-methylcytosine (5mC) (**A**) and 6mA (**B**) containing double-stranded oligonucleotide substrates. The levels of 5mC and its oxidized products (5-hydroxymethylcytosine: 5hmC and 5fC) are represented relative to the 5mC level at t=0. The levels of 6mA are represented relative to 6mA level at t=0. Error bars denote standard deviations from three independent experiments.

The online version of this article includes the following figure supplement(s) for figure 3:

**Figure supplement 1.** In vitro assays showing ten-eleven translocation (TET) activity profile on double-stranded oligonucleotide substrates containing or not 5-methylcytosine (5mC) (**A**) or N6-methyladenine (6mA) (**B**).

**Figure supplement 2.** Multiple sequence alignment of ten-eleven translocation (TET)/JBP family members.

---

(*Yao et al., 2018*; *Zhang et al., 2015*). However, both studies ran much longer reactions (up to 10 hr) and used different sources of recombinant protein (*Drosophila* TET catalytic domain purified from human HEK293T cells). Notably, *Zhang et al., 2015* only found around 2.5% of 6mA demethylation at 30 min and less than 25% after 10 hr of incubation as measured by HPLC-MS/MS analyses. These results suggest that *Drosophila* TET may oxidize 6mA, but with a much lower affinity than 5mC since we observed a near complete oxidation of 5mC after 1 min. and no significant decrease in 6mA levels after 30 min. of reaction (for identical concentrations of substrate and enzyme). It is possible too that the preparation of TET catalytic domain in different systems changes its enzymatic activity, potentially in relation to distinct post-translational modifications. Still, it is worth mentioning that a distant TET homolog in the fungus *Coprinopsis cinerea* was recently shown to oxidize both 5mC and 6mA (*Mu et al., 2022*). Importantly, its peculiar capacity to bind and demethylate 6mA requires key residues within its catalytic domain which are not conserved in other TET homologs including in *Drosophila* (*Figure 3—figure supplement 2*). These observations support our conclusion that *Drosophila* TET does not serve as 6mA demethylase.

Although previous reports suggested that TET controls fly viability, ovarian development, or adult brain formation by demethylating 6mA (*Yao et al., 2018*; *Zhang et al., 2015*), the functional importance of TET enzymatic activity has never been tested genetically as available *tet* alleles either abolish its expression or delete the whole catalytic domain. In view of our results and to address this issue, we generated a catalytic dead mutant allele of *tet* (*tet^{CD}*). Accordingly, the conserved HxD iron-binding motif required for the catalytic activity of TET/AlkB dioxygenase family of enzymes (*Hu et al., 2013*; *Tahiliani et al., 2009*; *Xu and Bochtler, 2020*) was mutated by CRISPR/Cas9-mediated homologous recombination using a *tet-GFP* knock-in which allows to tag all protein TET isoforms with GFP (*Figure 4A and B*). Of note, immunostaining in the larval CNS confirmed that TET is widely expressed in this tissue and showed that the H1947Y/D1949A mutation does not alter TET expression or its nuclear localization (*Figure 4C–H*). Strikingly, while *tet^{null}* homozygote individuals die at the pupal stage (*Delatte et al., 2016*), we found that *tet^{CD/CD}* as well as *tet^{CD/null}* pupae had a normal hatching rate and gave rise to viable adult flies (*Figure 4I*). We did not observe the lethality of *tet^{CD/CD}* individuals at earlier developmental stages either (*Figure 4—figure supplement 1*). Then, we assessed whether this mutation affected adult wing positioning, ovarian development, or mushroom body formation as reported upon TET loss of expression (*Wang et al., 2018*; *Yao et al., 2018*; *Zhang et al., 2015*). Yet, *tet^{CD/CD}* flies did not exhibit the 'held out' wing positioning defects present in *tet^{DMAD1/DMAD2}* adult escapers (*Figure 4J, J' and J''*). Similarly, atrophied ovaries or mushroom body projection defects

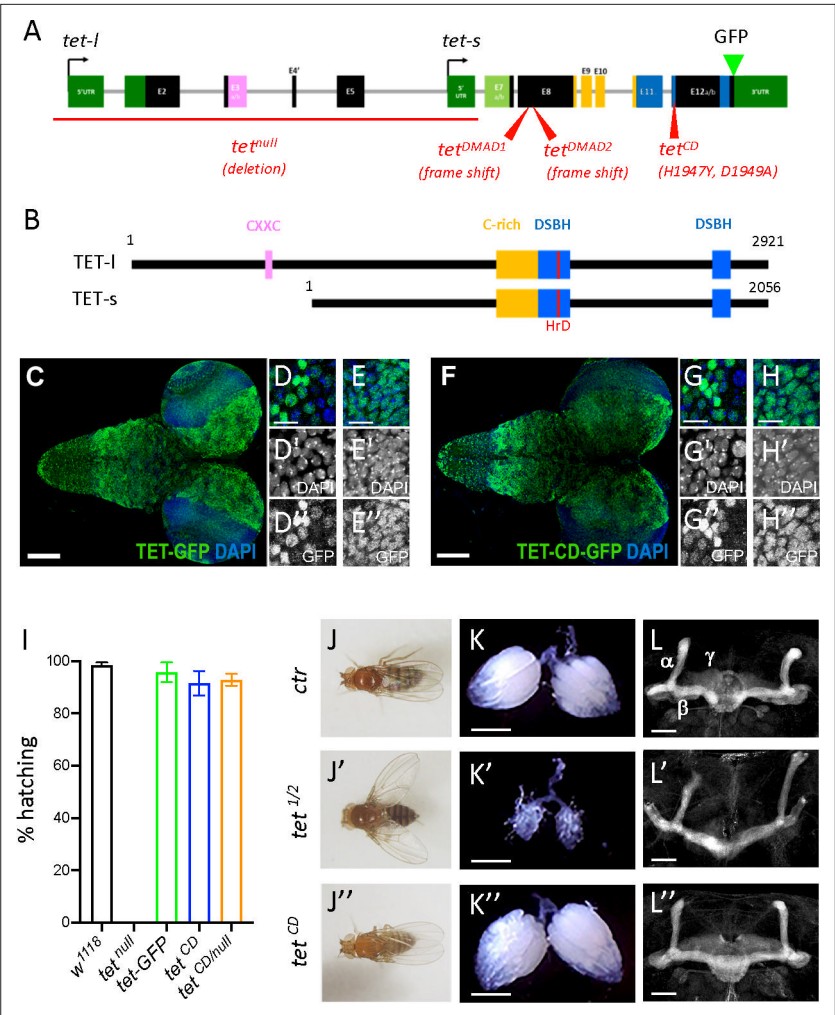

**Figure 4.** Ten-eleven translocation (TET) catalytic activity is largely dispensable in *Drosophila*. (**A, B**) Schematic representation of *tet* locus (**A**) and main protein isoforms (**B**). (**A**) *tet* is transcribed from two alternative promoters giving rise to *tet-long (tet-l)* and *tet-short (tet-s)* isoforms. Filled boxes represent exons; non-coding exons (UTR) are depicted in green, and coding exons in black or according to their domain-associated color. Introns are represented as gray lines (not to scale). The *tet null, DMAD1, DMAD2,* and *catalytic dead (CD)* alleles are depicted in red. The location of the GFP insertion generated by CRISRP/Cas9-mediated knock-in is also indicated. (**B**) The conserved domains of TET are colored; pink: CXXC DNA binding domain, orange: Cystein-rich domain, blue: double-stranded ß helix (DSBH) domain, red: HxD (iron binding motif). Amino acid positions are indicated according to the longest TET-l and TET-s isoforms. (**C–H**) Expression pattern of the wild-type and catalytic-dead versions of TET in the larval central nervous system (CNS). *tet-GFP* (**C–E**) and *tet^CD^-GFP* (**F–H**) knock-in lines were used to detect TET proteins by confocal imaging after immunostaining against GFP (green). Nuclei were labeled with DAPI (blue). (**C, F**): stitched images showing dorsal views of the entire CNS. Scale bar: 100 µm. (**D, E, G, H**): high-magnification views of TET expression in the ventral nerve cord (**D, G**) or the central brain (**E, H**). DAPI-only and GFP-only channels are presented in the middle (') and lower ('') panels, respectively. Scale bar: 10 µm. (**i**) Percentage of adult flies of the indicated genotypes hatching from their pupal case. Means and standard deviations from four independent experiments. (**J–L**) Wing positioning (**J**), ovaries (**K**), and mushroom bodies (**L**) of wild-type adult flies (*ctr: tet-GFP*) as compared to flies lacking TET expression (*tet^1/2^: tet^DMAD1/DMAD2^* adult escapers) or TET enzymatic activity (*tet^CD^*). (**L-L"**) Immunostaining against Fas2 on adult brains was used to label mushroom body α, β, and γ lobes. (**K-K"**) Scale bar 500 µm. (**L-L"**) Scale bar 50 µm.

The online version of this article includes the following figure supplement(s) for figure 4:

**Figure supplement 1.** Survival assays shows the percentage of hatching embryos, larvae, and adults of the indicated genotypes.

observed in the absence of TET expression were not reproduced when only its catalytic activity was impaired (*Figure 4K, K', K'' , L and L', L''*). Thus, TET function in *Drosophila* seems essentially independent of its enzymatic activity, indicating that TET-mediated regulation of 6mA level, if it truly happens, is not essential either for fly development.

## Conclusions

In sum, our results confirm that 6mA is present only at very low levels in the *Drosophila* genome. With only a few hundred methylated adenines per haplogenome, we argue that 6mA is unlikely to play a major regulatory function in normal conditions. In addition, we did not find any evidence that TET loss promotes 6mA accumulation. Rather, our results strongly suggest that this conserved enzyme is not a methyladenine demethylase and that its catalytic activity is largely dispensable for *Drosophila* development. Further experiments will be necessary to firmly establish whether regulated adenine methylation/demethylation takes place and what are the enzymes involved, not only in *Drosophila* but also in other metazoans. Besides, our analyses call for further investigations of the molecular mechanisms underlying the essential, enzymatic-independent, mode of action of TET.

## Methods
### Fly strains and breeding

The following *D. melanogaster* strains were used: *w^1118* (control, Bloomington), *tet^null* (*Delatte et al., 2016*), *tet^DMAD1*, *tet^DMAD2* (*Zhang et al., 2015*). *tet^null* (a kind gift from Dr R. Steward) was generated by FRT site recombination between two PBac insertions, resulting in the absence of *tet* transcription (*Delatte et al., 2016*). *tet^DMAD1* and *tet^DMAD2* (kind gifts from Dr. D. Chen) were obtained by CRISPR and produce truncated TET proteins deleted from their C-terminal domain (including the whole catalytic domain) (*Zhang et al., 2015*). The *tet-GFP* knock-in line was generated by InDroso Functional Genomics (Rennes, France) using CRISPR/Cas9-mediated homologous recombination to insert the EGFP in the frame with the last amino acid of TET. Similarly, the catalytic dead *tet^CD* flies were generated by CRISPR/Cas9-mediated homologous recombination in the *tet-GFP* background to mutate the TET HRD motif into YRA (*dm6:* chromosome 3 L:2,791,624 'A' to 'C' and 3 L:2,791,631 'C' to 'T'). In both cases, the resulting flies were validated by sequencing.

Unless otherwise specified, stock maintenance and sample collection were performed using classic fly medium (75 g/l organic corn flour, 28 g/l dry yeast, 40 g/l sucrose, 8 g/l agar, 10 ml/l Moldex 20%) with a 12 hr dark:light cycle. Germ-free *Drosophila* lines were generated as described (*Sabat et al., 2015*). Briefly: embryos were collected on grape juice agar plates, dechorionated with 2.7% bleach for 2–3 min, washed in sterile ddH2O, and transferred to standard fly medium supplemented with antibiotics (50 µg/ml amoxicillin, tetracyclin, kanamycin and puromycin) for at least two successive generations. When 'holidic' medium was used to avoid any source of contamination by exogenous DNA, embryos from germ-free adults were collected on grape juice agar plates, dechorionated, washed with ddH2O, and transferred to a chemically-defined medium, using the amino acid ratio of the FLYAA recipe (*Piper et al., 2017*), together with antibiotics to maintain axenic conditions. All crosses and larvae collections were performed at 25 °C.

### Viability assays

Embryos were collected at 25 °C from 1-week-old flies over 8 h on grape juice agar plates. For each genotype, batches of 100 embryos were transferred to corn flour-yeast-agar plates; the number of first instar larvae was counted after 30 hr, the number of pupae was counted at day 9 and the number of hatched adults was counted from days 10–15. Each experiment was repeated at least four times.

### Immunostainings

Third instar larvae or adult fly brains were dissected in 1 X phosphate buffer saline (PBS) and fixed for 25 min in PBS containing 4% paraformaldehyde. Fixed samples were washed rapidly twice with PBS and three times for 15 min with PBS-0.3% Triton X-100 (PBT) before being pre-incubated for 1 h in PBT-1% bovine serum Albumin (BSA, Sigma). Samples were incubated overnight at 4 °C with primary antibody diluted in PBT-1% BSA, washed three times for 15 min in PBT, and incubated with respective secondary antibodies diluted in PBT-1% BSA for 3 hr at room temperature or overnight at

4 °C. Samples were washed in PBT and mounted in Vectashield-DAPI (Vector Laboratories). Images were acquired using a Leica LSM800 confocal microscope. The following antibodies were used: goat anti-GFP (Abcam, 1/500), mouse anti-Fasciclin II (DSHB, 1/25), donkey anti-goat Alexa Fluor 488 (Invitrogen, 1/1000), donkey anti-mouse Cy3 (Jackson Immuno, 1/1000).

## Bacterial contamination assays

The presence of bacterial DNA contamination in parental and 'germ-free' derived stocks was checked by PCR using universal primers targeting bacterial *16 S rDNA* (*16* S-s: 5'-AGAGTTTGATCCTGGC TCAG-3', *16 S-r:* 5'-GGTTACCTTGTTACGACTT-3' *Weisburg et al., 1991*) and primers amplifying the *wsp* gene from the endosymbiont *Wolbachia* (*wsp*-s: 5'-TGGTCCAATAAGTGATGAAGAAAC-3', *wsp*-r: 5'-AAAAATTAAACGCTACTCCA-3' as from *Jeyaprakash and Hoy, 2000*). The presence of any contaminant was also checked by DNA sequencing: for each sample, genomic DNA from 10 adult flies (five males/five females) was extracted using DNeasy Blood & Tissue Kit (Qiagen). DNA was resuspended and sheared in 1 X TE (0.1 mM EDTA, 10 mM Tris HCl pH 8.0) by sonication for 20 min (30 sec ON/30 sec OFF) using the Bioruptor Pico (Diagenode) to obtain an average size of ~300 pb. DNA libraries were prepared from 1 µg DNA using the NEBNext Ultra II DNA Library Prep Kit (Illumina) following manufacturer's instructions and sequencing was performed by Novogene (Cambridge, UK) using NovaSeq 6000 (paired-end, 150pb). Between 15–19 million reads were obtained per sample. To assess the presence of contamination, the resulting reads were first aligned to the *Drosophila* reference genome (dm6 Ensembl release 70) with Bowtie2. Unaligned reads were then processed for blast search to bacteria, viral, and fungal genomes using the DecontaMiner tool (*Sangiovanni et al., 2019*).

## LC-MS/MS analyses

For DNA purification, whole larvae, bleach-dechorionated embryos, or dissected adult brains of the required genotypes were washed in sterile PBS, crushed in lysis buffer (100 mM Tris-HCl pH 9, 100 mM EDTA, 1% SDS), incubated at 70 °C for 30 min and then in 1 M potassium acetate at 4 °C for 30 min. After centrifugation at 12,000 g for 20 min, the supernatant was digested with RNAse A and RNAse H for 3 hr at 37 °C, extracted twice with phenol-chloroform-isoamyl alcohol (25:24:1) and precipitated with isopropanol. DNA from dissected third instar larval brains (around 100 per sample) was extracted using DNeasy Blood & Tissue Kit (Qiagen), digested with RNAse A and RNAse H for 3 hr. In both types of extraction, DNA was precipitated with 500 mM ammonium acetate (pH 6.2). DNA pellets were dissolved in ddH2O, and their concentration, as the absence of RNA contamination, was checked on a Qubit 3.0 fluorometer (Invitrogen).

Up to 1200 ng of DNA per sample were digested to nucleosides using 0.6 U nuclease P1 from *P. citrinum* (Sigma-Aldrich), 0.2 U snake venom phosphodiesterase from C. *adamanteus* (Worthington), 0.2 U bovine intestine phosphatase (Sigma-Aldrich), 10 U benzonase (Sigma-Aldrich), 200 ng Pentostatin (Sigma-Aldrich), and 500 ng Tetrahydrouridine (Merck-Millipore) in 5 mM Tris (pH 8) and 1 mM MgCl2 for 2 hr at 37 °C. 1000 ng of digested DNA were spiked with internal standard ($D_3$-5mC and $D_2$,$^{15}N_2$-5hmC, 250 fmol each) and subjected to analysis by LC-MS (Agilent 1260 Infinity system in combination with an Agilent 6470 Triple Quadrupole mass spectrometer equipped with an electrospray ion source (ESI)). The solvents consisted of 5 mM ammonium acetate buffer (pH 5.3, adjusted with acetic acid; solvent A) and LC-MS grade acetonitrile (solvent B; Honeywell). A C18 reverse HPLC column (SynergiTM 4 µM particle size, 80 Å pore size, 250 × 2.0 mm; Phenomenex) was used at a temperature of 35 °C and a constant flow rate of 0.5 mL/min was applied. The compounds were eluted with a linear gradient of 0–20% solvent B over 10 min. Initial conditions were regenerated with 100% solvent A for 5 min. The four main nucleosides were detected photometrically at 254 nm *via* a diode array detector (DAD). The following ESI parameters were used: gas temperature 300 °C, gas flow 7 L/min, nebulizer pressure 60 psi, sheath gas temperature 400 °C, sheath gas flow 12 L/min, capillary voltage 3000 V, nozzle voltage 0 V. The MS was operated in the positive ion mode using Agilent MassHunter software in the MRM (multiple reaction monitoring) mode. Therefore, the following mass transitions were used to detect the respective modifications: 6mA 266->150; 5mC 242->126; $D_3$-5mC 245->129; 5hmC 258->142; $D_2$,$^{15}N_2$-5hmC 262->146; 5fC 256->140. For absolute quantification, internal and external calibrations were applied as described previously (*Kellner et al., 2014*), except for 6mA and 5fC, for which only external calibration was performed.

## SMRT sequencing analyses

Around 250 brains from *w^1118* or *tet^null* third instar larvae were dissected for each sample. Tissues were crushed in lysis buffer (100 mM Tris-HCl pH 9, 100 mM EDTA, 100 mM NaCl,1% SDS), digested with RNase A for 15 min at room temperature and 30 min at 65 °C, before being incubated with four volumes of 3.2 M LiCl, 0.9 M KAc for 30 min at 4 °C. After centrifugation at 12,000 g for 20 min at 4 °C, the supernatant was extracted twice with phenol-chloroform-isoamyl alcohol (25:24:1) and DNA was precipitated with isopropanol. 5 µg of DNA were used to prepare each sequencing library.

SMRT-seq was performed at the Gentyane Sequencing Platform (Clermont-Ferrand, France) with a PacBio Sequel Sequencer (Pacific Biosciences, Menlo Park, CA, USA). The SMRTBell libraries were prepared using a SMRTbell Express 2 Template prep kit, following the manufacturer's recommendations. High Molecular Weight Genomic DNA (5 µg) was sheared with the 40 kb program using a Diagenode Megaruptor (Diagenode) to generate DNA fragments of approximately 30 kb. Assessment of the fragment size distribution was performed with a Femto Pulse (Agilent Technologies, Santa Clara, CA, USA). Sheared genomic DNA was carried into enzymatic reactions to remove single-strand overhangs and to repair any damage that may be present on the DNA backbone. An A-tailing reaction followed by the overhang adapter ligation was conducted to generate the SMRTBell templates. After a 0.45 X AMPure PB beads purification, the samples were size-selected using the BluePippin (Sage Science, Beverly, MA, USA) to recover all the material above 15 kb. The samples were then purified with 0.45 X AMPure PB Beads to obtain the final libraries of around 30 kb. The SMRTBell libraries were quality inspected and quantified on a Femto Pulse and a Qubit fluorimeter with Qubit dsDNA HS reagent Assay kit (Life Technologies). A ready-to-sequence SMRTBell Polymerase Complex was created using a Binding Kit 3.0 (PacBio) and the primer V4, the diffusion loading protocol was used, according to the manufacturer's instructions. The PacBio Sequel instrument was programmed to load a 6 pM library and samples were sequenced on PacBio SMRTCells v2.0 (Pacific Biosciences), acquiring one movie of 600 min per SMRTcell.

For each sequenced sample, SMRT-seq reads were aligned using pbmm2 tool (https://github.com/PacificBiosciences/pbmm2) (version 1.1.0) on *Drosophila* genome (dm6 Ensembl release 70). For each condition (wild-type or *tet^null*), a fusion of alignments of the three biological replicates was done using « samtools merge » (*Li et al., 2009*). 6mA detection was performed on individual and on merged samples with IpdSummary tool from KineticTools (http://github.com/PacificBiosciences/kineticsTools) (version 2.4.1) applying the following parameters: `--identify m6A --numWorkers 16 --p-value 0.01 --identifyMinCov 5 –methylFraction`. To detect 6mA with higher confidence, we applied several thresholds on coverage and modificationQV (QV) with homemade scripts in bash and R. The SMRT-seq data are deposited under GEO accession number GSE206852.

## Purification of *Drosophila* TET catalytic domain

The catalytic domain of *drosophila* TET (dTET) was cloned in pET28a expression vector. The His-tagged protein was overexpressed in *E. coli* BL21 (DE3) CodonPlus RIL cells for 17 h at 16 °C. Cells were resuspended in lysis buffer (50 mM HEPES pH 7.5, 20 mM imidazole, 500 mM NaCl, 1 mM DTT, 10% glycerol, and supplemented with protease inhibitor 0.2 mM PMSF) and disrupted using Bandelin Sonoplus ultrasonic homogenizer. The cell lysates were cleared by centrifugation (Lynx 600 (Thermo), Fiberlite F21−8x50 y) at 38.300 g for 30 min at 4 °C and the supernatant was loaded onto an affinity column packed with Ni-NTA agarose beads (Genaxxon, Germany). The column was washed with wash buffer (50 mM HEPES pH 7.5, 20 mM imidazole, 500 mM NaCl, 1 mM DTT, 10% glycerol) then the recombinant protein was eluted with elution buffer (50 mM HEPES pH 7.5, 250 mM imidazole, 500 mM NaCl, 1 mM DTT, 10% glycerol). Purified protein was dialyzed against dialysis buffer I (50 mM HEPES pH 7.5, 1 mM DTT, 300 mM NaCl, and 10% glycerol) followed by dialysis buffer II (50 mM HEPES pH 7.5, 1 mM DTT, 300 mM NaCl, and 50% glycerol).

## TET activity assays

The double-stranded DNA substrates were prepared by annealing forward oligos and reverse complement counterparts by heating at 95 °C followed by bringing the temperature to RT slowly on the heat block in Annealing Buffer (10 mM Tris-HCl pH7.5 100 mM NaCl). Forward oligos sequences containing a 5mC or 6mA modified nucleotide were the following: 5mC 5'-GTAAGTCTGGCA5mCGTGAGCCTCAGAG-3', 6mA 5'-GTAAGTCTGGCG6mAGTGAGCCTCAGAG-3'. The reaction was performed with

0.5 µM DNA substrate and 2 µM recombinant dTET in Reaction Buffer (50 mM HEPES pH 6.8, 100 µM Ammonium ion(II) sulfate hexahydrate, 1 mM, α-ketoglutarate, 1 mM ascorbic acid, and 150 mM NaCl) at 37 °C. Reaction was stopped at different time points by adding 2 µl of 0.5 M EDTA to a 40 µl volume reaction followed by heating at 90 °C for 5 min. Samples were treated with proteinase K for 1 hr at 50 °C and precipitated with ethanol. The level of 6mA, 5mC, 5hmC, and 5fC was analyzed by LC-MS/MS as described above.

## Acknowledgements

We are grateful to Dr. A Molaro for the critical reading of the manuscript and to the members of the iGReD for helpful discussions. We also thank Dr. V Gautier and the Gentyane platform at the Clermont-Ferrand INRAE for PacBio sequencing and the iGReD CLIC imaging facility for help with confocal experiments. We thank the *Drosophila* Bloomington Stock Center, Dr. R Steward (Piscataway), and Dr. D Chen (Beijing) for fly stocks, as well as the Developmental Study Hybridoma Bank for antibodies. This project was supported by grants from the Agence Nationale de la Recherche (ANR-17-CE12-0030-03), Fondation ARC (PJA20171206371), and I-Site Cap20-25 (EpiMob) to LW. MB was supported by fellowships from the Université Clermont-Auvergne and the Fondation pour la Recherche Médicale (FRM). GG was supported by fellowships from the Université Clermont-Auvergne and the Ligue Nationale Contre le Cancer. MH was funded by the Deutsche Forschungsgemeinschaft (DFG, German Research Foundation) TRR-319 TP C03, SPP1784, HE 3397/13–2, and HE 3397/14–2.

## Additional information

### Funding

| Funder | Grant reference number | Author |
| --- | --- | --- |
| Agence Nationale de la Recherche | ANR-17-CE12-0030-03 | Lucas Waltzer |
| Fondation ARC pour la Recherche sur le Cancer | PJA20171206371 | Lucas Waltzer |
| Clermont Université | i-Site CAP20-25 | Lucas Waltzer |
| Fondation pour la Recherche Médicale | | Manon Boulet |
| Ligue Contre le Cancer | | Guerric Gilbert |
| Deutsche Forschungsgemeinschaft | SPP1784 | Mark Helm |
| Deutsche Forschungsgemeinschaft | TRR-319 TP C03 | Mark Helm |
| Deutsche Forschungsgemeinschaft | HE 3397/13-2 | Mark Helm |
| Deutsche Forschungsgemeinschaft | HE 3397/14-2 | Mark Helm |

The funders had no role in study design, data collection and interpretation, or the decision to submit the work for publication.

### Author contributions

Manon Boulet, Guerric Gilbert, Conceptualization, Investigation, Writing – review and editing; Yoan Renaud, Formal analysis, Investigation, Visualization, Writing – review and editing; Martina Schmidt-Dengler, Investigation, Writing – review and editing; Emilie Plantié, Romane Bertrand, Xinsheng Nan, Investigation; Tomasz Jurkowski, Supervision, Writing – review and editing; Mark Helm, Supervision, Funding acquisition, Writing – review and editing; Laurence Vandel, Lucas Waltzer, Conceptualization, Supervision, Funding acquisition, Writing – original draft, Writing – review and editing

**Author ORCIDs**
Manon Boulet ⓘ https://orcid.org/0000-0003-1308-3303
Yoan Renaud ⓘ https://orcid.org/0000-0002-4036-8315
Xinsheng Nan ⓘ http://orcid.org/0000-0002-0865-7934
Laurence Vandel ⓘ https://orcid.org/0000-0002-3692-2942
Lucas Waltzer ⓘ https://orcid.org/0000-0002-5361-727X

Reviewer #1 (Public Review) https://doi.org/10.7554/eLife.91655.3.sa1
Reviewer #2 (Public Review) https://doi.org/10.7554/eLife.91655.3.sa2
Author Response https://doi.org/10.7554/eLife.91655.3.sa3

## Additional files

### Supplementary files

• Supplementary file 1. Genome coverages of wild-type and *tet*<sup>null</sup> fusin datasets in SMRT-seq.

• Supplementary file 2. Percentages of 6mA/A according to QV and coverage cut-off in wild-type and tet*null* fusion datasets.

• Supplementary file 3. Proportions of replicated 6mA (triplicated and/or duplicated).

• Supplementary file 4. Percentages of 6mA/A and 6mA fractions of methylation according to QV in the three wild-type or three *tet*<sup>null</sup> datasets (coverage ≥25 x).

• MDAR checklist

### Data availability

The SMRT-seq data have been deposited on GEO (GSE206852). Source data files have been provided for all images of gels in main and supplementary figures. All other data generated during this study are included in the manuscript and supporting files.

The following dataset was generated:

| Author(s) | Year | Dataset title | Dataset URL | Database and Identifier |
|---|---|---|---|---|
| Waltzer L, Renaud Y | 2022 | Analysis of gene expression in the *Drosophila* larval central nervous system | https://www.ncbi.nlm.nih.gov/geo/query/acc.cgi?acc=GSE206852 | NCBI Gene Expression Omnibus, GSE206852 |

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
