## [Editor Report · eLife assessment]

This study investigates the presence of DNA adenine methylation (6mA) and the associated function of TET enzyme, a DNA methylation mark eraser, in *Drosophila*. The study presents **valuable** findings on the scarcity of 6mA in the *Drosophila* genome and challenges previous findings regarding the role of TET in 6mA modification. The evidence supporting the claims is **solid**, and the paper has the potential to stimulate re-evaluations of the significance and regulatory mechanisms of 6mA DNA modifications in *Drosophila*.

---

## [Referee Report · Reviewer #1 (Public Review)]

This work challenges previously published results regarding the presence and abundance of 6mA in *Drosophila* genome, as well as the claim that the TET or DMAD enzyme serves as the ‘eraser’ of this DNA methylation mark and its roles in development. This information is needed to clarify these questions in the field. Generally speaking, the methods for fly husbandry and treatment seem to be in accordance with those established ones in the field.

Here are a couple of suggestions that could be discussed with the current work and addressed in the future, in order to better understand the roles of 6mA and TET.

1. Regarding the estimated ‘200 to 400 methylated adenines per haplogenome’, some insights regarding where they are enriched in the genome could inform the potential target sites regulated by 6mA.

2. The TET-GFP and TET-CD-GFP knock-in lines give proper nuclear localization and could be used to identify genomic regions bound with full-length TET and TET-CD using anti-GFP for ChIP-seq or CUT&RUN (or CUT&TAG).

---

## [Referee Report · Reviewer #2 (Public Review)]

DNA adenine methylation (6mA) is a rediscovered modification that has been described in a wide range of eukaryotes. However, 6mA presence in eukaryote remains controversial due to low abundance of its modification in eukaryotic genome. In this manuscript, Boulet et al. re-investigate 6mA presence in *Drosophila* using axenic or conventional fly to avoid contaminant from feeding bacteria. By using these flies, they find that 6mA is rare but present in drosophila genome by performing LC/MS/MS. They also find that the loss of TET (also known as DMAD) does not impact on 6mA levels in drosophila, contrary to previous studies. In addition, the authors find that TET is required for fly development in its enzymatic activity-independent manner.

The strength of this study is, compared to previous studies of 6mA in *Drosophila*, the authors employ axenic or conventional fly for 6mA analysis. These fly strains make it possible to analyze 6mA presence in drosophila without bacterial contaminant. This established method is valuable in this field.

---

## [Author Response]

The following is the authors’ response to the original reviews.

**Reviewer #1 (Public Review):**
This work challenges previously published results regarding the presence and abundance of 6mA in the *Drosophila* genome, as well as the claim that the TET or DMAD enzyme serves as the ‘eraser’ of this DNA methylation mark and its roles in development. This information is needed to clarify these questions in the field. I am less familiar with the biochemical approaches in this work, so my comments are mainly on the genetic analyses. Generally speaking, the methods for fly husbandry and treatment seem to be in accordance with those established in the field.

Response : We thank the reviewer for his/her work and positive assessment of our manuscript.

**Reviewer #2 (Public Review):**
DNA adenine methylation (6mA) is a rediscovered modification that has been described in a wide range of eukaryotes. However, 6mA presence in eukaryote remains controversial due to the low abundance of its modification in eukaryotic genome. In this manuscript, Boulet et al. re-investigate 6mA presence in drosophila using axenic or conventional fly to avoid contaminants from feeding bacteria. By using these flies, they find that 6mA is rare but present in the *Drosophila* genome by performing LC/MS/MS. They also find that the loss of TET (also known as DMAD) does not impact 6mA levels in drosophila, contrary to previous studies. In addition, the authors find that TET is required for fly development in its enzymatic activity-independent manner.The strength of this study is, that compared to previous studies of 6mA in *Drosophila*, the authors employed axenic or conventional fly for 6mA analysis. These fly strains make it possible to analyze 6mA presence in drosophila without bacterial contaminant. Therefore, showing data of 6mA abundance in drosophila by performing LC-MS/MS in this manuscript is more convincing as compared with previous studies. Intriguingly, the authors find that the conserved iron-binding motif required for the catalytic activity of TET is dispensable for its function. This finding could be important to reveal TET function in organisms whose genomic 5mC levels are very low.The manuscript in this paper is well written but some aspects of data analysis and discussion need to be clarified and extended.1. It is convincing that an increase in 6mA levels is not observed in TETnull presented in Fig1. But it seems 6mA levels are altered in Ax.TET1/2 compared with Ax.TETwt and Ax.TETnull presented in Fig1f (and also WT vs TET1/2 presented in Fig1g). Is it sure that no statistically significant were not observed between Ax.TET1/2 and Ax.TETwt?1. The representing data of in vitro demethylation assay presented in Fig.3 is convincing, but it is not well discussed and analyzed why these results are contrary to previous reports (Yao et al., 2018 and Zhang et al., 2015).

We thank the reviewer for his/her work and positive assessment of our manuscript.

(1) We repeated our statistical analyses and confirmed that there is no significant difference between wildtype and tet1/2 mutant embryos in axenic conditions (Welch two sample t-test : p=0.075).

(2) We added some elements in the revised manuscript to discuss the possible reasons for the discrepancies with previous reports. Notably both studies performed the in vitro demethylation assays over a much longer time course and with different sources of recombinant proteins. Zhang et al. purified TET catalytic domain from human cells (HEK293T) and observed around 2.5% of 6mA demethylation at 30 min and less than 25% after 10 hours of incubation as measured by HPLC-MS/MS analyses. Yao et al. incubated recombinant TET catalytic domain with 6mA DNA for 3h and observed a 25% decrease in 6mA levels as measured by dot blot. These results suggest that *Drosophila* TET may oxidize 6mA, but with a much lower affinity than 5mC since with observed a near complete oxidation of 5mC after 1 minute and no decrease in 6mA levels after 30 minutes of reaction (for identical concentrations of substrate and enzyme). It is possible too that the preparation of TET catalytic domain in different systems changes its enzymatic activity, potentially in relation with distinct post-translational modifications. Still, as already mentioned in our manuscript, extensive biochemical analyses of the distant TET homolog from the fungus Coprinopsis cinerea (Mu et al., Nature Chem Biol 2022) strongly argue that TET enzymes do not harbor the residues required to serve as 6mA demethylase.

**Reviewer #1 (Recommendations For The Authors):**
Here are one comment (#1) and a couple of questions (#2-3) that could be addressed in the future, in order to understand the roles of 6mA and TET. Even though #2 and #3 are likely beyond the scope of this paper, #1 should be addressed within the scope of this work and compared with previous reports.1. The phenotypic analyses in Fig. 4 should use tet_null/Deficiency and tet_CD/Deficiency for their potential phenotypes. This needs to be addressed since both the tet_null and the tet_CD were generated using the same starting fly line (GFP knock-in). Using a deficiency chromosome and testing these alleles in hemizygotes would be helpful to eliminate any secondary effects due to genetic background issues.

Thanks for this comment. Actually, tet_null and tet_CD were not generated using the same starting lines. Whereas tet_cd was generated (by CRISPR) using the tet-GFP knock-in line, tet_null was generated by FRT site recombination between two PBac insertions (Delatte et al. 2016). As for tet1 and tet2 (used in allelic combination in Fig 4 J-L), they correspond to two distinct mutant alleles generated by CRISPR (Zhang et al. 2015). We have clarified this in the M&M (page 9).

1. Regarding the estimated ‘200 to 400 methylated adenines per haplogenome’, is there any insight into where are they located in the genome?

It is an interesting question and we initially used SMRT-seq sequencing to obtain this kind of information. As it turned out that this technique gives a high level of false positive, we should consider with caution the interpretation of these data and we decided not to include them in the manuscript. Still, we characterized the genomic features of the 6mA detected using stringent criteria (mQV>100, cov>25x in the fusion dataset and triplicated across samples of the same genotype). Both in wild type and tet_null, 6mA were dispersed along each chromosome although few of them were found on chromosome X. In both cases there appeared to be a higher accumulation of 6mAs on the histone locus and the transposon-rich tip of chromosome X, but 6mA density remained below 1.3/kb in other genomic regions. Comparisons with annotated genomic regions indicated that 6mA were enriched in long interspersed nuclear elements (LINEs) and satellite repeats, and depleted in 3’UTR and exons, but there was no significant difference in their repartition between the two genetic contexts. Besides, motif analyses showed similar enrichments in both conditions, with GAG triplet accounting for more than one quarter of all the sites. Whether this reflects the specificity of a putative adenine methylase or a technical bias associated the with SMTR-seq technology remains to be established.

1. The TET-GFP and TET-CD-GFP knock-in lines give proper nuclear localization and could be used to identify genomic regions bound with full-length TET and TET-CD using anti-GFP for ChIP-seq or CUT&RUN (or CUT&TAG).

Indeed, this is a line of research that we are following up and will be part of another study. Actually, our ChIP-seq experiments indicate that they bind on the same genomic regions.

**Reviewer #2 (Recommendations For The Authors):**
I think the major findings of this paper are showing 6mA present in *Drosophila* by using xenic or conventional breeding conditions and finding that TET function independently of its catalytic activity is essential for fly development. The authors could have been more precise in title and abstract to emphasize these findings.

We have now modified the abstract to try to emphasize these findings.

The authors claim that any increase of 6mA levels was not observed in both TETnull and TET1/2, but it is not sufficiently convincing. Because it seems 6mA levels were increased in Ax. tet1/2 embryo as compared with in Ax.wt embryo (Fig.1). In this scenario, 6mA abundance in both TETnull and TET1/2 mutant are supposed to be the same. It would be better to re-analyze data carefully and discuss if 6mA levels were significantly increased in TET1/2, and why 6mA levels are different between TETnull and TET1/2. Additionally, the authors describe that the TET null mutant is pupal lethal, while the TET1/2 survivor is available. The text suggests that TET1/2 could have partial functionality on fly development (Fig.4). It would be better to check whether the N-terminus of TET is expressed in the TET1/2 mutant.

Indeed, the increase in 6mA levels in Ax. tet1/2 embryo seems consequent (although it is not statistically significant) and no increase was observed in Ax tet_null embryos. Thus, the putative effect on 6mA levels in tet1/2 embryos may not be directly due to the absence of TET function. We now mention in the revised manuscript (page 6) that ‘the apparent increase in 6mA levels in tet1/2 axenic embryos was not reproduced in tet_null embryos, suggesting that it does not simply reflect the tet loss of function, and that it was not statistically significant’. Besides, we do not have an antibody to check whether the N-terminus of TET is expressed in the tet1/2 mutants, but the western blot published by Zhang et al 2015 shows that tet2 mutation leads to the expression of TET N-terminal domain. This N-terminal domain could have partial TET functionality and/or interfere with the function of other factors (notably those implicated in 6mA metabolism).

The authors show that SMRT-seq data did not reveal an increase in 6mA levels in loss of TET (Fig.2). It is convincing that total 6mA abundance was not altered by loss of TET. But were 6mA-accumulated locus/regions observed in WT not altered by loss of TET?

Please refer to our answer to reviewer 1 on that point.

It remains unclear that the TET proteins the authors prepared do not exhibit 6mA demethylate activity in vitro, contrary to what was reported in previous papers (Fig.3). I think the preparation of recombinant proteins may make different results between this and previous papers. Yao et al., 2018 and Zhang et al., 2015 used recombinant proteins purified from Human cells or insect cells, while the author purified them from *E. coli*. Additionally, it's mentioned that VK Rao et al., 2020 demonstrated cdk5-mediated phosphorylation of Tet3 increases its in catalytic activity in vitro. These previous reports suggest modification of TET could change demethylase activity. More analysis and discussion are needed to support the conclusion.

Thanks for your insights. This in an important point and we added the following elements in the revised manuscript to discuss possible reasons for the discrepancies with previous reports (pages 7-8): ‘Our results contrast with previous reports showing that recombinant *Drosophila* TET demethylates 6mA on dsDNA in vitro (Yao et al. 2018; Zhang et al., 2015a). However, both studies ran much longer reactions (up to 10 hours) and used different sources of recombinant protein (*Drosophila* TET catalytic domain purified from human HEK293T cells). Notably, Zhang et al. (2015a) only found around 2.5% of 6mA demethylation at 30 min and less than 25% after 10 hours of incubation as measured by HPLC-MS/MS analyses. These results suggest that *Drosophila* TET may oxidize 6mA, but with a much lower affinity than 5mC since with observed a near complete oxidation of 5mC after 1 min. and no significant decrease in 6mA levels after 30 min. of reaction (for identical concentrations of substrate and enzyme). It is possible too that the preparation of TET catalytic domain in different systems changes its enzymatic activity, potentially in relation to distinct post-translational modifications.’